# Application of Computing as a High-Practicability and -Efficiency Auxiliary Tool in Nanodrugs Discovery

**DOI:** 10.3390/pharmaceutics15041064

**Published:** 2023-03-25

**Authors:** Ke Xu, Shilin Li, Yangkai Zhou, Xinglong Gao, Jie Mei, Ying Liu

**Affiliations:** 1CAS Key Laboratory for Biomedical Effects of Nanomaterials and Nanosafety, CAS Center for Excellence in Nanoscience, National Center for Nanoscience and Technology of China, Beijing 100190, China; 2University of Chinese Academy of Sciences, Beijing 100049, China; 3GBA National Institute for Nanotechnology Innovation, Guangzhou 510700, China

**Keywords:** nanodrug, drug research and development, computing, model prediction, molecular simulation

## Abstract

Research and development (R&D) of nanodrugs is a long, complex and uncertain process. Since the 1960s, computing has been used as an auxiliary tool in the field of drug discovery. Many cases have proven the practicability and efficiency of computing in drug discovery. Over the past decade, computing, especially model prediction and molecular simulation, has been gradually applied to nanodrug R&D, providing substantive solutions to many problems. Computing has made important contributions to promoting data-driven decision-making and reducing failure rates and time costs in discovery and development of nanodrugs. However, there are still a few articles to examine, and it is necessary to summarize the development of the research direction. In the review, we summarize application of computing in various stages of nanodrug R&D, including physicochemical properties and biological activities prediction, pharmacokinetics analysis, toxicological assessment and other related applications. Moreover, current challenges and future perspectives of the computing methods are also discussed, with a view to help computing become a high-practicability and -efficiency auxiliary tool in nanodrugs discovery and development.

## 1. Introduction

It is estimated that the price of developing a therapeutic drug is USD 2.6 billion, and it will take about 10 years, with a clinical success rate of less than 10% [1]. It can be said that R&D of new drugs is a time-consuming, costly and high-risk process. Generally, the R&D process of a new drug includes the following stages: drug discovery and pre-study, preclinical research, IND application, clinical research, NDA application and marketing monitoring (Figure 1). As a new therapeutic agent, a nanodrug is a kind of pharmaceutical preparation with nanometer-scale and obvious nanoscale effects developed by nanotechnology. According to the different characteristics of various nanodrugs, they can be roughly divided into three categories, including drugs with direct nanolization, nano-carrier drugs and other nanodrugs (e.g., protein nanodrugs and nanobody drugs). Compared with traditional small-molecule drugs, nanodrugs have many characteristics, such as small particles, large specific surface area, high surface reactivity, many active centers and strong adsorption capacity [2]. Using nanomaterials as drug carriers can improve absorption and utilization of drugs, achieve efficient target delivery, prolong half-life of drug consumption, enhance immune response and reduce harmful side effects on normal tissues [2,3,4,5,6,7], which makes it possible to create more efficient, less toxic and more intelligent therapies for medical diseases. Although many new nanodrugs and drug delivery systems have been reported continuously, few of them can be truly transformed into clinical applications. Due to special physical, chemical and biological characteristics, nanodrugs have attracted much attention, which leads to a relatively high threshold for their R&D and production to be approved. Since 1989, only 78 nanodrugs have been approved and entered the global market, and lipids and polymers are still the main delivery carriers of nanodrugs. At present, nanodrugs still have problems, such as poor stability in vivo and rapid circulation and metabolism, which largely limits efficacy of nanodrugs. In addition, R&D of nanodrugs also faces many difficulties, such as high costs, insufficient understanding of biological mechanisms and poor experimental repeatability. Therefore, R&D of nanodrugs urgently needs efficient and low-cost auxiliary tools to speed up its process.

With rapid development of computer technology, as a powerful tool, it has been successfully applied in artificial intelligence, life science, materials science and many other fields and has achieved remarkable results. Taking the pharmaceutical field as an example, researchers from Insilico Medicine developed an artificial intelligence model (named GENTRL) and used it to design the potential molecular structure of DDR1 kinase inhibitor within 21 days and completed a preliminary biological validation within 43 days [8]. In addition, researchers at the University of Southern California’s Dornseff College of Arts and Sciences have developed a new drug virtual-screening method (called V-SYNTHES), which can screen billions of compounds in a faster and cheaper way to find new and targeted drug therapies [9]. These methods can complete the work that traditional methods need to take months to years to complete in a relatively short time, which greatly saves the time required for drug R&D and reduces the high cost.

According to PubMed statistics, from 2010 to 2022, the number and proportion of published articles related to computing in the nano-field showed an upward trend (Figure 2). Similarly, computing has also been widely used in nanodrug R&D. For example, according to quantitative structure–activity relationship models for nanomaterials (Nano-QSAR), Puzyn et al. [10] built a model by using the multiple regression method combined with a genetic algorithm (GA-MLR) to accurately predict the cytotoxicity (EC_50_) of 17 different types of metal oxide nanoparticles to bacteria Escherichia coli. Walkey et al. [11] constructed a multivariable machine learning model based on the information obtained from experiments on protein crowns adsorbed on nanomaterials to predict in vitro cell uptake of nanomaterials. Fourches et al. [12] used the descriptors calculated by small-molecular ligands modified on the surface of carbon nanotubes (CNTs) to build a quantitative structure–activity relationship (QSAR) model to predict protein adsorption and cytotoxicity in CNTs. Nagpal et al. [13] developed a pharmacokinetic model of polymer nano-biopharmaceutical preparations based on the physiological mechanism. A smart differential evolution algorithm was applied to optimize the existing clinical data of polymer nano-biopharmaceutical preparations for the model. In addition to simulating the key parameters of the pharmacokinetic behaviors of nano-biopharmaceutical preparations with different polymer formulations in the human body, the model can also be used to calculate release rate of polymer nano-biopharmaceutical preparations in vivo. Chew et al. [14] used molecular dynamics to simulate nanoparticles to obtain simulation-derived descriptors and then constructed a quantitative nanostructure–activity relationship (QNAR) model to predict the physicochemical and biological properties of nanoparticles. It can be seen from these cases that computing has been applied in most stages of the nanodrug R&D process, and the computational methods involved can be divided into two categories, including model prediction and molecular simulation (Figure 3).

Compared with the experimental method, the model prediction can make qualitative and quantitative prediction on the materials that have not been used or even synthesized based on the existing data, provide reasonable prediction results for the unknown NM and significantly reduce costs and use of animals (Figure 3a). At present, in the research of prediction and evaluation of nanodrug properties, the widely used computational models mainly include the machine learning model, quantitative structure–activity relationship (QSAR) model and PBPK model. In addition, due to the multi-component and complex microstructure of drug carriers and the limitations of experimental conditions, it is usually difficult to observe the microscopic morphology and distribution of nanoparticles, which is not conducive to further research. The molecular simulation method can effectively overcome this shortcoming, which can be used to simulate the molecular structure and behavior with the molecular model at the atomic level and then simulate the various physicochemical properties of the molecular system, providing a more reliable direction for further research of the nanodrug delivery system (Figure 3b). The main steps of molecular simulation include selecting the real system, establishing the model, running the simulation and analyzing the data. According to different simulation scales, there are four main methods of molecular simulation, including quantum mechanics, molecular mechanics, molecular dynamics and Monte Carlo methods.

With this background, the high-speed processing and distributed computing capabilities of computer technology can be used to simulate the information processing and learning capabilities of the human brain so that computer technology can analyze and solve the problems faced in the process of nanodrug research and development, which may provide new opportunities for design and development of nanodrugs, accelerate the R&D process and reduce the failure rate. Therefore, this review will systematically introduce application of computing in regulation and design of nanodrugs from the perspective of algorithm prediction and molecular simulation and briefly summarize the challenges faced in the application process.

## 2. Computing in Prediction of Physicochemical Properties

The physicochemical properties (size, shape, surface chemistry, elasticity, etc.) of drugs will be significantly changed after nanometerization (including direct nanosizing of drugs and nano-carrier-based drug delivery systems). Studies have shown that physicochemical properties are the key factors for the fate of nanodrugs in vivo. These properties have important effects on cell uptake, endocytosis mechanism, distribution of living tissue, pharmacokinetics in vivo, immune response and biological safety [15,16,17,18]. Therefore, better understanding and accurate prediction of physicochemical properties are more conducive to design of nano-pharmaceutical systems with required pharmacokinetic and pharmacodynamic characteristics so as to achieve efficient delivery of nanodrugs and improve drug utilization.

### 2.1. Computing in Prediction of Hydrophilic–Hydrophobic Property

The hydrophilic and hydrophobic properties of nanodrugs reflect their hydrophilic or lipophilic properties. Studies have shown that the physicochemical properties of nanodrugs are highly related to many biological effects, such as cell uptake, in vivo distribution, protein adsorption, pharmacokinetics, etc. In general, hydrophobic nanoparticles are more conducive to cell uptake and can absorb more plasma proteins [17,19]. It can be said that hydrophilic–hydrophobic property is an important prerequisite and indicator for evaluating the biological effects of nanodrugs. Therefore, how to quickly and accurately obtain hydrophilic–hydrophobic property of nanodrugs is of great significance to guide rational design of nanodrugs. Hydrophilic–hydrophobic property of nanodrugs can be quantitatively characterized by oil–water partition coefficient (measured by shaking flask method, chromatography or production column method), hydrophobic probe molecular method and surface tension method. At present, most researchers use logP to express hydrophilic–hydrophobic property of nanodrugs. The traditional computing methods for predicting hydrophilic–hydrophobic include atomic group method and group contribution method [20,21], but the models based on these methods are often too complex and have poor generalization.

The classic quantitative structure–activity relationship (QSAR), as a quantitative description and study of the relationship between the structure and activity of compounds, can be used to build a prediction model by using mathematical statistical methods and then predict various properties of compounds [22]. As a statistical simulation method, the quantitative structure–activity relationship (QSAR) method has the advantages of small computation and good prediction ability and has been used to evaluate the properties of nanomaterials. Puzyn et al. [23] called it the “Nano-QSAR” model.

At present, some methods based on quantitative structure–activity relationship (QSAR) have been used to predict hydrophilic–hydrophobic property of nanomaterials. For example, Wang et al. [24] established a quantitative nanostructure–activity relationship (QNAR) model to predict hydrophilic–hydrophobic property of gold nanoparticles using the virtual gold nanoparticles (vGNP) library based on experimental results and a large number of accurately calculated nanodescriptors.

However, selection and determination of molecular descriptors is a very important link in QSAR modeling research [25]. Due to the lack of appropriate nano-descriptors for nanomaterials, applicability and predictability of the QSAR model are seriously limited [26,27]. In view of this, Chew et al. [14] used molecular dynamics to simulate nanoparticles to obtain simulation-derived descriptors and then constructed a quantitative nanostructure–activity relationship (QNAR) model to predict hydrophilic–hydrophobic property of nanoparticles (Figure 4).

In addition, more and more machine learning algorithms have also been used to build models to predict hydrophilic–hydrophobic property. For example, inspired by face recognition technology, Yan et al. [28] developed a novel nanostructure labeling method that automatically converts nanostructures into images for convolution neural network modeling and successfully predicts hydrophilic–hydrophobic property of nanoparticles (Figure 4b,c).

### 2.2. Computing in Prediction of Surface Charge

Due to the nanoscale size of nanodrugs, their surface charge will significantly affect their aggregation performance and stability. Generally, the more charges on the surface of nanodrugs, the more repulsive their particles will be, thus achieving stability of the whole system. In addition, surface charge of nanodrugs also plays an important role in determining the fate of the body [37,38]. Surface charge of nanodrugs is generally evaluated based on Zeta potential. Surface potential is closely related to particle size, composition and dispersion medium of nanodrugs. The measured value depends on the measurement conditions, such as dispersion medium, ion concentration, pH and instrument parameters, and it is necessary to select appropriate methods and media for research, such as phase analysis light scattering (PALS), electrophoretic light scattering (ELS) or tunable resistance pulse sensing (TRPS). For the large and complex nanomaterial system, it is obviously time-consuming and labor-intensive to evaluate surface charge through experimental methods. However, using computational methods to predict Zeta potential of nanomaterials can greatly accelerate evaluation and screening of nanomaterials.

Except for the previously mentioned surface charge prediction method for nanomaterials [14,28], Mikolajczyk et al. [34] reported a machine learning model for predicting Zeta potential of 15 metal oxide nanoparticles using 11 image-based descriptors and 17 computational descriptors. They used the linear regression method to predict Zeta potential of nanomaterials. The RMSE of the test set reached 1.25 mV, and the *Q*_2EXT_ was 0.87. Sizochenko et al. [35] constructed the measurement dataset containing the Zeta potential of metal oxide nanoparticles under different environmental conditions and developed the nanomaterial structure–property prediction model (nano-SPR), which can quantitatively describe the relationship between the structural characteristics of 208 metal oxide nanoparticles in different biological media and their Zeta potential using neural network algorithm. The authors also optimized the model by changing the super parameters of the neural network (such as number of hidden layers, activation function and iteration number), and the prediction accuracy of the final model can reach 76.25%.

### 2.3. Computing in Prediction of Other Physicochemical Properties

In addition to the above two important physical and chemical factors, other properties, including particle size, shape, elasticity and porosity, also affect uptake and release of nanodrugs and are closely related to ADMET events in vivo [39,40]. For example, Sonavane et al. [41] compared the blood circulation and biological distribution of gold nanoparticles with different sizes (15, 50, 100 and 200 nm). They found that smaller nanoparticles (15 and 50 nm) showed longer cycle times and higher accumulation in all organs. In particular, they can even pass the blood–brain barrier. However, larger nanoparticles (100 and 200 nm) showed shorter cycle time and more accumulation in liver and spleen. Therefore, predicting these properties is very important to evaluate efficacy, safety and metabolism of nanodrugs. Recently, He et al. [31] used LightGBM method to predict the particle size and polydispersity coefficient of nanocrystals under different preparation methods. The research results confirmed that the prediction model has good prediction accuracy for nanocrystals prepared by ”top-down” method.

Overall, prediction of physicochemical properties of nanomaterials by means of computational tools is relatively extensive. Application of AI has also accelerated development of high-throughput screening of nanomaterials, which is critical to find suitable carrier materials for developing nanodrug delivery systems. In addition, the physical and chemical properties of nanodrugs, such as particle size, shape, hydrophobicity and surface charge, have a crucial impact on ADMET properties and cytological effects. Precise prediction and regulation of physical and chemical properties can greatly optimize biological characteristics of nanodrugs in vivo and tap their greater potential. For example, adjusting the different physical and chemical properties of nanomaterials can regulate the immune system of the body and contribute to immunotherapy [18]. Currently, how to obtain an efficient and safe optimal combination, so that the physical and chemical properties are not simply added but give full play to the synergy to achieve the effect of “1 + 1 > 2”, is still a challenge. At the same time, most AI algorithms applied for prediction of physicochemical properties are relatively simple and their ability to learn the data characteristics of nanomaterials is still lacking, which leads to poor applicability of constructed models, and predicted properties mainly focus on particle size, shape, hydrophilic–hydrophobic property and surface charge. Prediction of other properties (such as porosity, hardness and mechanical properties) is still less studied, and there is a lack of universal model that can predict multiple properties at the same time. However, with the emergence of new algorithms, increase in publicly available data and deepening of nanodrug research, many of the above problems will be smoothly solved. It will also be possible to carry out reasonable control and design on the physicochemical properties of nanodrugs through computing so as to realize personalized nanodrug delivery system.

## 3. Computing in Prediction of Biological Activities

### 3.1. Computing in Prediction of Cellular Uptake

Cellular uptake of drugs is one of the important factors that determine efficacy of drugs. Nanoparticles enter the cell mainly through endocytosis [42]. More and more evidence shows that cell uptake is an important basis for biological effects of nanodrugs and plays a decisive role in bioavailability and efficacy of drugs targeted in cells. However, uptake of nanoparticles is a complex process, which is usually affected by many factors, such as particle concentration, particle size, particle morphology, particle surface modification, cell type and time [43]. At present, the quantitative analysis methods for uptake of nanodrugs by cells studied in vitro mainly include flow cytometry (FCM), laser confocal microscopy (CLSM), scanning transmission electron microscopy (STEM) and inductively coupled plasma mass spectrometry (ICP-MS). However, these determination methods are complex and cumbersome, which is not conducive to rapid evaluation of nanodrugs. In order to alleviate the above problems, it can be regarded as an efficient way to dynamically and accurately evaluate uptake of nanodrugs through the calculation method of model prediction.

Apart from some of the above QSAR models and deep learning methods used to predict cell uptake [14,24,28], as early as 2012, Ghorbanzadeh et al. [36] predicted the uptake behavior of 109 magnetic fluorescent nanomaterials by pancreatic cancer cells (PaCa_2_) based on multiple linear regression (MLR) and multi-layer perceptual neural network (MLP-NN). The R of the prediction results of the two models are 0.769 and 0.934, and the RMSE is 0.364 and 0.150, respectively. The prediction performance of the neural network is significantly improved compared with the traditional MLR. The sensitivity analysis of MLP-NN model showed that the number of hydrogen donors in the organic coating of nanomaterials was the main factor affecting cell uptake. In addition, Luan et al. [29] developed a quantitative nanostructure activity relationship (QNAR) model through linear multiple linear regression (MLR) and nonlinear artificial neural network (ANN) technology, which can accurately predict the cell uptake value of pancreatic cancer cells to nanoparticles. At the same time, the model also has good predictability to external datasets. Ali et al. [30] constructed five different convolutional neural network models to predict uptake of TNBC cells to nanoparticles loaded with anticancer drugs. The model showed high accuracy and had potential for cell uptake assessment at the early stage of drug development.

### 3.2. Computing in Prediction of Protein Adsorption

After a nanodrug enters the biological environment, the endogenous proteins will rapidly bind to the surface of the nanoparticles to form a protein coronal structure [44,45]. Protein corona can reshape the physical and chemical properties (such as size, hydrophilic–hydrophobic, surface charge and stability) of the nanodrugs connected to it and may adversely affect the affinity, structure and function of proteins, thus masking recognition of nanoparticles [43,46,47,48]. Therefore, the study of the protein adsorbed in the corona phase will be able to better predict the biological characteristics of nanodrugs. At present, the largest technical difficulty in protein crown research is proteome analysis, which usually requires highly sensitive protein mass spectrometry. The protein crown composition refers to the relative protein abundance (RPA) of the total protein in the crown, which is an important parameter to describe the protein crown. However, the experimental method is not only time-consuming but also expensive. Therefore, predicting the composition of protein crowns by computation rather than laboratory measurement can greatly save resources and even predict the unknown interactions between biological entities and various nanodrugs. However, due to the complex properties of nanomaterials and the complex composition of protein crowns, it is still challenging to predict the protein crowns and their mediated nanobiological effects.

In addition to the above methods used to predict protein adsorption [28], Ban et al. [32] built a small protein crown database of nanomaterial. Through optimizing machine learning using random forest algorithm combined with the internal data of the database and the double verification of cell experiments, they achieved accurate prediction of single protein and key functional proteins (such as immune protein, complement protein and apolipoprotein) in the complex protein crown of nanomaterial, as well as cell recognition mediated by functional protein crown (such as cell uptake and cytokine release). Compared with the fitting results of quantitative factors and target values obtained by traditional linear regression model (most R^2^ < 0.4), the R^2^ of this model can reach more than 0.75, and its robustness, universality and accuracy are improved after optimization. Analysis of the formation mechanism of protein crown will help to design ideal and safe nanomaterials in the fields of nanomedicine, biosensors and organ targeting. In addition, Ouassil et al. [33] developed a random forest classifier using mass spectrometry data to predict protein adsorption on single-wall carbon nanotubes (SWCNTs) functionalized by ssDNA, with accuracy of 78%, AUC of 76%, precision of 70% and recall of 65%. The machine learning model allows people to quickly analyze the protein attributes in the public database to determine the protein characteristics and interested proteins of the crown of SWCNTs.

Overall, biological activities of nanodrug will affect release, transport and metabolism of drugs in vivo. It is of great significance to fully understand the biological activities of drugs for assessment of their non-clinical safety and effectiveness. At present, the biological activity prediction research of nanodrug carriers is still less, including cell uptake, protein adsorption and oxidative stress (mainly causing toxic effects; see Section 5), which is largely due to the more complex biological activities, insufficient research on the mechanism of action and immature characterization methods compared with physicochemical properties. In addition, traditional ML algorithms are often difficult to meet the needs of biological activity prediction, and there is still upside potential in algorithm improvement. In addition to model prediction, molecular simulation is also a promising means. The combination of model prediction and molecular simulation is expected to realize prediction of the dynamic changes of nanodrugs in the biological environment, which will greatly promote the research of biological activities of nanodrugs and is a direction worthy of exploration in the future.

## 4. Computing in Prediction of ADME

As a new drug, nanodrugs have the characteristics of slow controlled release and targeting compared with ordinary drugs and can improve drug bioavailability. However, potential toxicity and unclear mechanism of action have been the important reasons for the high failure rate of nanodrugs in clinical development [49]. Obviously, understanding the fate (absorption, distribution, metabolism and excretion) of nanomedicine in vivo is of great significance for evaluating the overall safety and promoting development of nanomedicine (Figure 5a,b) [50]. As a simulation tool, the physiology-based pharmacokinetic (PBPK) model can be used to quantitatively describe and predict concentration, time distribution and exposure of drugs in blood and individual organs, which is crucial for efficacy/toxicity prediction and risk assessment and is helpful to guide design of nanodrugs and optimize their PK. Since the first semi-PBPK model of liposome doxorubicin was published in 1999 [51], PBPK models have been studied for many types of nanoparticles. However, applications to nanoparticles are still very limited and challenging due to the complex in vivo transport mechanisms of nanoparticles, such as opsonization and mononuclear phagocyte system (MPS) uptake, enhanced permeability and retention (EPR) effect, lymphatic transport, cellular recognition and internalization, enzymatic degradation and physical property changes [52]. At present, PBPK models developed for nanoparticles are basically based on blood circulation, which connects organs or tissues into a system. Different from the PBPK model structure of small-molecular drugs and biological products, the unique interaction between nanoparticles and physiological system should also be considered when constructing the PBPK model for nanoparticles.

### 4.1. Computing in Prediction of Absorption

Nanodrugs can enter the body through many ways, such as vein, oral, subcutaneous or muscle [55,56]. The route of administration is an important factor that determines the absorption of nanodrugs. After intravenous administration, nanodrugs directly enter the systemic circulation. After oral administration, drug-loaded particles can be absorbed in the gastrointestinal tract through paracellular transport, cell ingestion and M cell uptake [57], and a small amount may be absorbed into the systemic circulation through the lymphatic system. After subcutaneous and muscular administration, the drug-loaded particles are absorbed mainly through macrophages and lymph uptake [58] and then distributed into the systemic circulation. Compared with the in vivo absorption of ordinary drugs, which is mainly reflected by the determination of the concentration of active drugs in the systemic circulation, evaluation of the in vivo absorption of carrier nanodrugs is relatively complex. Therefore, PBPK models for conventional molecules are generally not suitable for simulating absorption of nanodrugs [59].

In fact, nanoparticles enter cells mainly through endocytosis [42,43,60]. However, endocytosis of nanoparticles is a very complex process and simulation of endocytosis of nanoparticles increases the complexity and uncertainty of the PBPK model. To avoid this complexity, PBPK model can use linear equations to simulate the size independent endocytosis of nanoparticles. For example, Bachler et al. [61] have based uptake rates on organ-specific characteristics, including capillary wall type, phagocytosis efficiency as well as the amount of Ag NPs that passes through the capillary walls of each organ as determined from organ blood flow normalized to the total blood volume. However, the uptake rate of nanoparticles in vivo is not linear. It has been observed that the uptake rate will slow down as saturation approaches [62]. In order to predict endocytosis of nanomaterials more accurately, Lin et al. [63] used the Hill function (a time-dependent uptake rate function) to predict the endocytosis of nanomaterials.

Furthermore, Rajoli et al. [64] applied PBPK modeling to address the feasibility of developing monthly intramuscular injectable nanodrugs for antiretrovirals. Based on the clinical data of each antiretroviral oral preparation, researchers expanded the PBPK model with an intramuscular depot compartment to simulate drug release and absorption. Subsequently, the full PBPK model was verified against the existing antiretroviral nanodrugs. Through simulation, the authors were able to optimize the combination of dose and release rate of eight antiretroviral drugs to maintain therapeutic plasma concentration during the entire administration interval. The prediction results show that, when the optimized dose is within the dose range of intramuscular injection, the feasibility of each antiretroviral drug nanodrug per month is confirmed.

### 4.2. Computing in Prediction of Distribution

After absorption, the nanodrug will be distributed throughout the body and in the tissue [65,66,67,68]. The specific distribution still depends on the physical and chemical properties and surface properties of the drug-loaded particles and is also affected by many factors, such as blood protein binding, tissue and organ hemodynamics and vascular tissue morphology (such as the size of the gap). Because nanoparticles are often used to change the biological distribution of encapsulated drugs, the method of accurately predicting the distribution of nanoparticles in vivo will greatly guide the design and optimization of nanodrugs. Traditionally, two distribution models have been proposed to describe the tissue distribution dynamics of small molecules: blood perfusion limited model and diffusion limited model [14]. However, these two distribution models are not sufficient to accurately describe the complex tissue distribution process of biological products and nanoparticles. Therefore, it is very important to develop a PBPK model suitable for evaluation of nanodrug distribution in vivo.

For example, Li et al. [69] applied a general PBPK model with limited permeability to fit the biological distribution data of five poly(lactic-co-glycolic) acid (PLGA) nanoparticle formulations prepared with varied content of monomethoxypoly (ethyleneglycol) (mPEG). Then, multiple linear regression was performed to establish the relationship between the properties of nanoparticles (size, Zeta potential and number of PEG molecules per unit surface area) and biological distribution parameters. For all five initial formulations, the developed model fully simulated the experimental data. The predicted biological distribution curve based on the physical and chemical properties of PLGA-mPEG495 nanoparticles (a sixth formulation) is close to the experimental data, reflecting properly developed property–biodistribution relationships, which indicates that the model is suitable for describing the biological distribution of PLGA-mPEG nanoparticles.

In addition, Bachler et al. [61] used PBPK model to estimate the absorption and systemic distribution of silver nanoparticles and ionic silver after skin, intestinal or lung exposure. This model can describe the tissue distribution of ionic silver and silver nanoparticles with the size of 15 nm to 150 nm in rats and humans for risk assessment. In particular, the model can distinguish the mucociliary cleared from the bile particles cleared in the feces. Similarly, Li et al. [70] established a similar PBPK model, including mucociliary clearance, phagocytosis and entry into the systemic circulation by alveolar wall penetration, to study the biological distribution dynamics of combustion-generated CeO_2_ nanoparticles inhaled by rats. The model successfully predicted the biological distribution of CeO_2_ nanoparticles in various organs and showed that most nanoparticles were captured by phagocytes.

Unlike the conventional methods, Bachler et al. [54] proposed a new two-step method to evaluate the biodynamics of inhaled NPs. In a first step, alveolar epithelial cellular monolayers (CMLs) at the air–liquid interface (ALI) were exposed to aerosolized NPs to determine their translocation kinetics across the epithelial tissue barrier. In a second step, the distribution to secondary organs was predicted with a physiologically based pharmacokinetic (PBPK) model (Figure 5c). The experimental results show that the combination of in vitro experiments and computational methods can accurately predict the pharmacokinetics of NPs in vivo (Figure 5d) and has the potential to replace short-term animal studies aimed at evaluating the lung absorption and biological distribution of NPs.

### 4.3. Computing in Prediction of Metabolism

The active drugs and depolymerized carrier materials in the nano-carrier drugs are mainly metabolized by metabolic enzymes in liver and other tissues in vivo. In addition, drug-loaded particles are easy to be swallowed by MPS and then degraded or metabolized by lysosomes, which may affect the type and quantity of metabolites of drugs and carrier materials. Therefore, it is important to determine the main metabolic pathways of active drugs and carrier materials and analyze their metabolites. However, in the published PBPK model, due to the lack of knowledge of chemical and metabolic degradation processes, first-order degradation kinetics are usually assumed for most nanoparticles. For example, Lin et al. [71] assumed that all Quantum Dot 705 (QD705) disposition was associated with the first-order rate of metabolism, kf, in the liver, and the time-dependent nature of this rate constant was described by a Hill function.

Due to the unique physical and chemical properties of nanomaterials and the complexity of biological effects, it is difficult to directly measure the metabolic data of nanodrugs in vivo using existing technologies. One solution is that, as in Lin et al. [71], the degradation rate measured in vitro can be used as the initial value of the parameters of the metabolic model and the initial value can be further optimized by fitting the model with the available ADME data in vivo [59].

### 4.4. Computing in Prediction of Excretion

Generally, nanodrugs are excreted mainly through glomerular filtration into urine or excreted with feces in the form of bile secretion through the liver [52,72,73,74]. Due to the influence of different physicochemical properties, the drug excretion pathway and excretion rate will vary after the administration of nanodrugs. It has been proven that the renal clearance depends on the size of nanomaterials, where nanomaterials less than 5.5 nm can be effectively and completely eliminated by the renal clearance [72], while the renal clearance of larger nanomaterials is very slow [75]. Aborig et al. [76] assumed that the renal excretion of gold nanoparticles with diameter larger than 10 nm was negligible. In addition, hepatocytes are conducive to uptake of positively charged nanoparticles and intact nanoparticles or their degradation products can be excreted in bile, but the hepatobiliary excretion of nanoparticles is usually very slow (from hours to months) [74]. As excretion is an important process to reduce the potential toxicity of nanodrugs, it is extremely necessary to develop an appropriate PBPK model to evaluate it for nanodrug R&D.

At present, the PBPK model of nanomaterials mainly focuses on kidney and hepatobiliary clearance, and bile and urine excretion rates are highly influential parameters in many nanomaterial PBPK models [77]. Generally, these excretion parameters of nanomaterials can be mined from the literature. For example, according to the observation results in literature [78], in the PBPK model constructed by Carlander et al., the urine excretion of TiO_2_ NP was set to zero and the biokinetic distribution of TiO_2_ nanoparticles was accurately simulated [79]. Similarly, Lin et al. [77] used rates in their PBPK model for Au NPs, which were obtained from the study on tissue kinetics of PEG-coated Au NPs by Cho et al. [80].

Overall, the PBPK model is still the main application of computing in the study of nanodrug pharmacokinetics. Studying the pharmacokinetics of nanodrugs in vivo through calculation is conducive to evaluating safety and efficacy of drugs, guiding drug design, screening new drugs, optimizing drug delivery plan and significantly reducing time and costs in the research process. However, due to the lack of research on the metabolism process of nanomaterials in vivo, the PBPK model currently constructed mainly focuses on absorption, distribution and excretion of nanomaterials. In addition, due to the limited actual physiological data and lack of professionals, some parameters in many PBPK models still need to be further optimized and the prediction performance needs to be improved.

## 5. Computing in Prediction of Toxicological Assessment

Due to their special physicochemical characteristics, such as nanoscale effect and nanostructure effect, nanodrugs have relatively special biological characteristics. While providing benefits to clinical application, the safety risks they carry may also increase correspondingly [81]. In addition to intravenous injection, nanodrugs enter the human body mainly through skin penetration, the digestive system and respiratory system. In the blood, nanodrugs will interact with blood proteins to form protein crowns, which will affect the physicochemical properties of blood and cells [82]. In addition, nanodrugs can also reach various tissues of the human body through the blood circulation system, and some nanodrugs are absorbed by organs, which will cause damage to organs and tissues [83,84]. Toxicity has always been the focus of attention regarding safety of nanodrugs. At the same time, toxicity is also a major reason for failure of drug R&D. Therefore, only by accurately predicting toxicity of nanodrugs can we ensure safety of drugs, better protect patients and improve the efficiency of drug development.

At present, there are many kinds of nano-carrier materials with different toxicity mechanisms. In view of various mechanisms, although there have been a wide range of toxicological evaluation experiments and means of nanodrugs in the world [85,86], there has not yet been a unified framework to comprehensively and effectively evaluate toxicity of nanodrugs. Currently, to evaluate the toxicity of a nanodrug in detail, researchers usually need to evaluate it from many aspects, such as lung toxicity, kidney toxicity, immune toxicity, genetic toxicity, carcinogenicity and so on [87,88,89,90,91]. However, most of the traditional toxicological research methods are based on experimental tests, which have the problems of long time, high cost and cumbersome operation. Therefore, it is necessary to use non-test assessment methods in hazard assessment of nanodrugs. As an auxiliary tool for simulation and prediction, computing can make up for the above deficiencies to a large extent, identify and evaluate potential risks, reduce use of biochemical reagents and experimental animals, guide and optimize design of nanodrug structures and even explore new mechanisms of nanodrug toxicity.

In the past decade, many computational methods have been developed to predict toxicological effects in nano form (Table 1). Huang et al. [92] built a comprehensive dataset of 30 metal oxide nanoparticles (MeONPs) and generated descriptors of MeONPs through quantum chemical calculations. Based on the constructed database and quantum chemical calculation, the authors established a QSAR model using machine learning to predict the pulmonary inflammatory effect induced by MeONPs. The prediction accuracy of the model is more than 90%. Through verification of the experiment, it is proven that the external prediction ability of the model is good, and the accuracy rate is up to 86%. Machine learning model analysis shows that electronegativity, ζ-potential and cationic charge are the key parameters that affect the inflammatory effect induced by MeONPs. DFT calculation and experimental results further reveal the potential toxicity mechanism: metal oxides with low metal electronegativity and positive ζ-potential are more likely to cause lysosomal damage and inflammation. This work provides a tool for toxicity screening and safety design of nanomaterials and also provides the possibility of replacing large-scale animal experiments with computational simulation in the future. In addition to prediction of lung toxicity, Ban et al. [93] collected data from published literature to build a database containing 250 pieces of reproductive toxicity data of nanomaterials and used random forest algorithm to model and predict impact of nanomaterials on reproductive toxicity in mice. The results show that the RF model has good prediction performance in all subsets, with R^2^ values greater than 0.6. In addition, importance analysis of the characteristics of the random forest showed that type of nanomaterials (e.g., Ag, MWCNT and TiO_2_) and category of toxicity indicators (e.g., sperm parameters, testosterone level and testis index) were the most important factors to be considered when evaluating reproductive toxicity of nanomaterials.

Furthermore, Yu et al. [94] proposed a tree-based random forest feature importance and feature interaction network analysis framework (TBRFA) to predict and analyze pulmonary immune responses and lung burden of nanoparticles. The prediction accuracy of the model meets that the R^2^ value of all training sets is >0.9, and the R^2^ value of half of the test sets is >0.75. In addition, TBRFA uses multiway importance analysis to overcome the deviation caused by the structural imbalance of small sample datasets and identifies that exposure recovery time, dose, specific surface area of materials and material size are important factors that affect the biological effects induced by nanomaterials. TBRFA also builds feature interaction networks. By calculating the interaction coefficient between the two features, it revealed that the specific surface area and surface charge, specific surface area and length, length and diameter of materials play a role of mutual restriction and influence in the process of inducing biological effects and improved the interpretability of the model. This study provides an important idea for design and application of ideal nanoparticles.

Overall, toxicity determination is one of the most important and challenging steps in the drug discovery and development cycle. Similar to prediction of physicochemical properties, toxicity of nanodrugs and nanomaterials is mostly predicted by using ML algorithms and QSAR model. Use of predictive toxicology models provides a fast, cheap and reliable alternative for large-scale in vivo and in vitro bioassay, which helps to solve the ethical, economic and efficiency limitations of traditional toxicology experiments and also promotes understanding of toxicity mechanisms. However, the following challenges still exist in the prediction model of nanotoxicity. On the one hand, the amount of toxicity effect data of nanomaterials is limited, and most of them are in vitro data, lacking relevant data of toxicity in vivo. On the other hand, proper descriptors play a crucial role in the prediction performance of the models. At present, there is still a lack of descriptors that can accurately characterize the overall structure of nanomaterials.
pharmaceutics-15-01064-t001_Table 1Table 1Computational case for predicting nanotoxicity.Types of NanomaterialsDescriptorsComputing MethodsEnd Point of Toxic EffectReference**Metal and metal oxide nanoparticles**Physicochemical and 2D-topological descriptorsQSAR, ANNCC_50_, LC_50_, EC_50_[95]**Metal oxide nanoparticles**Periodic-table-based and physicochemical descriptorsANNlog(1/EC_50_)[96]**Metal oxide nanoparticles**Structural descriptorsQSAR, GA-MLRlog(1/EC_50_)[10]**carbon nanotubes**Molecular descriptorsQSAR, RF, kNN, SVMacute cytotoxicity[12]**Inorganic nanomaterials**Atom-based quantitative descriptorsLightGBMCytotoxicity[97]**TiO_2_ hybridized with multi-metallic (Ag, Au, Pt) alloy nanoparticles**Empirical descriptorsQSTR, MLR, KRR, SVR, GPR, RFREC_50_[98]**Metal oxide nanoparticles**Structural, periodic-table-based and physicochemical descriptorsC4.5, LGR, RF, kNN, DT, LWL, Bayesnet, SVMImmunotoxicity[99]**Two-dimensional nanomaterials**
Free energy analysis, MD, computational indicator of nanotoxicity (CIN2D)Cytotoxicity[100]**Silver nanoparticles**Physicochemical and experimental descriptorsDT, RFCytotoxicity[101]**Metal oxide nanoparticles**Periodic-table-based and physicochemical descriptorsLR, RF, SVM, NNCytotoxicity[102]**Multi-walled carbon nanotubes**Physicochemical and experimental descriptorsQSAR, PCA, LR, RF, SVM, NBGenotoxicity[103]**Metal oxide nanoparticles**Periodic-table-based and experimental descriptorsLDA, NB, MLogitR, SMO, AdaBoost, J48, RFEC_50_[104]**Cadmium-containing quantum dots and metal oxide nanoparticles**
LightGBMIC_50_[105]**Engineered nanomaterials**Physicochemical descriptorsRFDevelopmental toxicity[106]**TiO_2_ and ZnO nanoparticles**Physicochemical descriptorsMLR, LDALDH release[107]**Metal oxide nanoparticles**Molecular descriptorsQSAR, MLRlog(LC_50_)[108]**Metal oxide nanoparticles**SMILES-based optimal descriptorsQSARpEC_50_[109]**Metal oxide nanoparticles**SMILES-based optimal descriptorsQSAR, MC-PLSLC_50_[110]**Metal oxide nanoparticles**Physicochemical descriptorsQSAR, MLRlog(1/EC_50_)[111]AdaBoost: adaptive boosting, ANN: artificial neural network, C4.5: C4.5 decision tree, DT: decision table, GA-MLR: multiple regression method combined with genetic algorithm, GPR: Gaussian process regression, J48: C4.5 decision tree, kNN: k-nearest neighbor, KRR: kernel ridge regression, LDA: linear discriminant analysis, LGR: logistic regression, LightGBM: light gradient boosting machine, LR: logistic regression, LWL: locally weighted learning, MC-PLS: Monte Carlo partial least square, MD: molecular dynamic, MLogitR: multinomial logistic regression, MLR: multiple linear regression, NB: naive Bayes, NN: neural network, PCA: principal components analysis, QSAR: quantitative structure–activity relationship, QSTR: quantitative structure–toxicity relationship, RF: random forest, RFR: random forest regression, SMO: sequential minimal optimization, SVM: support vector machine, SVR: support vector regression.


## 6. Computing in Other Aspects of Nanodrug R&D Process

### 6.1. Computing in Prediction of Nanomodeling

Traditionally, the properties of nanomaterials are often measured and evaluated through experiments, but this is not only time-consuming and laborious but also may face ethical problems in animal experiments. In addition, changes in physicochemical properties of nanomaterials will also greatly affect their biological effects in vivo. For the same type of nanomaterials, it is often necessary to evaluate properties of different sizes and shapes. This has brought great challenges to the workload of the experiment and preparation process. Therefore, it is an economically feasible alternative to model for nanomaterials through computing method and then predict their properties.

For instance, Wang et al. [24] first synthesized a large library of gold nanoparticles and obtained comprehensive data on their characteristics and biological activities. At the same time, each nanoparticle in the library was simulated by calculating the nanostructure characteristics of each nanoparticle, and a virtual gold nanoparticle library was established using the in-house GNPrep program. In addition, a quantitative nanostructure–activity relationship (QNAR) model was built using the kNN algorithm to predict and design nanoparticles with required biological activities. The experimental results of the designed nanoparticles are consistent with the model predictions (Figure 6a). These findings show that it is completely feasible to predict the properties of nanomaterials through virtual modeling and can significantly reduce the efforts and costs of nanomaterial discovery. Further, Yan et al. [27] used the above GNPrep program to model for the real gold nanoparticles dataset and built a virtual gold nanoparticles library. Based on these vGNPs, the authors calculated new nano-descriptors and developed a quantitative nano-structure activity relationship (QNAR) model to predict the physicochemical properties and biological activities of gold nanoparticles. The prediction results show that the model has high predictability for physicochemical properties (log P and Zeta potential) and simple activities (two cellular uptakes) but moderate accuracy for more complex bioactivities (GNP-enzyme bindings and ROS induction). Similarly, Liu et al. [112] applied the above modeling strategy to build a virtual carbon nanoparticle library based on experimental data. They developed the QNAR model by calculating the nano-descriptors of the virtual carbon nanoparticles and successfully predicted the cytotoxicity and four different inflammatory reactions induced by the PM_2.5_ model.

### 6.2. Computing in High-Throughput Screening of Nanodrugs

With the increasing demand for nanodrugs, finding suitable new nanodrugs has become a major task in nanodrug research. However, the traditional drug screening mode is mainly based on the common cell model or animal model, which has limitations, such as high cost and long cycle, and high requirements on the operating skills of technical personnel [84]. It is difficult to carry out effective and economic screening of a certain number of samples in a short time, which greatly limits the development process of new nanodrugs. At present, computer-aided technologies (such as machine learning and molecular simulation) have been widely used in virtual screening of small-molecular drugs and have achieved impressive results. As an efficient and accurate means, computer-aided technology has also been used for high-throughput screening of nanodrugs to replace traditional methods.

Recently, some studies have shown that some anticancer drugs can form stable nanoparticles with ultra-high drug-loading volumes when combined with specific small-molecular excipients. In view of this, Reker et al. [115] combined molecular dynamics (MD) simulation and machine learning with a high-throughput experimental co-aggregation platform to quickly identify effective drug–excipient combinations. In this work, they extracted drugs and excipients from DrugBank and FDA libraries, respectively, and produced about 2.1 million drug–excipient combinations. Machine learning analysis of the dataset is completed using a random forest classifier. In addition, high-throughput dynamic light scattering (DLS) is also used to train and validate the platform. Although the relevant characteristics of MD contribute a great deal to accurate prediction of co-aggregation and help to understand stability of nanoparticles, considering the cost of calculation, it is obviously unrealistic to carry out MD for 2.1 million potential drug–excipient pairs. Therefore, researchers have applied machine learning models specifically for training of physicochemical properties. The improved model has slightly lower retrospective performance but higher computational ability and predicts formation of copolymer with a precision of 0.94. After that, several predicted condensates were verified in the test set through experiments, proving the reliable prediction ability of the screening platform.

Further, Zhu et al. [113] developed a drug screening system (called DeepScreen) utilizing convolutional neural network based on flow cytometry single-cell images. The model can efficiently extract, identify and locate tiny variation from cell apoptosis and slight changes in cellular period caused by drugs. Even under interference of nanomaterials and spontaneous fluorescence of drugs, the effect of drugs can be accurately judged by DeepScreen (Figure 6b). Compared with common experimental methods, using this model to screen nanodrugs can greatly shorten detection time from a few days to 2–6 h and significantly improve detection efficiency. Further, the experimental results show that the detection accuracy of the model is as high as 0.966. All these superior performances show that DeepScreen is a nanodrug screening system with broad application prospects.

### 6.3. Study on Mechanism of Action

When nanodrugs enter the body, they will interact with various components (such as cell membrane and protein) in the biological environment. Understanding how nanodrugs interact with biological components is crucial to grasp the absorption, distribution, metabolism and excretion process of nanodrugs in the body and clarify related risks to human health. Relatively speaking, due to the complexity of the behavior of nanodrugs in vivo, the existing in vitro experiments cannot completely simulate the internal environment. As a computational simulation tool, molecular simulation can simulate experiments that are difficult to carry out under normal conditions, reduce costs of experiments and also provide high security. Therefore, molecular simulation is a good alternative.

For example, Ding et al. [114] reported that virus-mimicking designer DNA nanostructures act like charge attraction on the interface of cell membrane. The results showed that cellular internalization of tetrahedral DNA nanostructures (TDNs) mainly depended on lipid mediation, where caveolin plays a key role in providing short-range attraction at the membrane interface. After that, the authors studied the interaction between a tetrahedral DNA nanostructure with edge length of 20 bp (TDN-20) and the simulated cell membrane by conducting dissipative particle dynamics (DPD) simulation (Figure 6c). Both simulation and experimental data show that TDN approaches the membrane mainly through its corners to minimize electrostatic repulsion, and they cause uneven charge redistribution in the membrane under the short-distance limit of caveolin. This work provides new insights into the mechanism of charge attraction at the nanoscale.

In addition, the method based on machine learning can also be used to predict the interaction sites of nanomaterials and proteins. Cha et al. [116] extended the machine learning algorithm for protein–protein interaction training to interaction between inorganic nanoparticles and proteins. The results showed that the predicted protein interaction sites of nanoparticles almost matched the experimental results. These findings can be extended to other organic and inorganic nanoparticles to predict their interactions with biomolecules and other chemical structures.

### 6.4. Guidance on Rational Design of Nanodrugs

Several studies have shown that the physicochemical properties of nanodrugs and nanomaterials will greatly affect their behavior and activities in vivo, such as cycle time, biological distribution, targeting and immunogenicity, and the physicochemical properties are relatively easy to be controlled and adjusted. Therefore, it is an excellent choice to design nanodrugs reasonably by adjusting physicochemical properties to build personalized functional nanodrugs.

For example, Shamay et al. [117] described a targeted drug delivery system that is accurately and quantitatively predicted to self-assemble into nanoparticles based on the molecular structures of drugs themselves. These drugs can be assembled with the help of sulfated indocyanines to form particles with up to 90% ultra-high drug loading. At the same time, the authors designed a quantitative structure–nanoparticle assembly prediction (QSNAP) model based on the decision tree algorithm, which can accurately predict nano-assembly and nanoparticle size. Based on the above model, the authors found targeted drug carrier nanoparticles (nanoparticles combined with kinase inhibitors sorafenib and trametinib) formed by self-assembly of small molecules. The nanoparticles can selectively target human colon cancer and liver cancer in in situ models expressing CAV1 to yield significant therapeutic effects while preventing inhibition of healthy tissues. This study provides guidance for computational design of nanodrugs based on the quantitative model of drug payload selection.

Furthermore, from bottom to top, it is also a good choice to develop nanodrugs with required functions based on the mechanism of action. However, there is still a lack of research on the in vivo behavior mechanism of nanodrugs and nanomaterials, which greatly increases blindness and uncertainty of nanodrug R&D. In view of this, Zhu et al. [118] combined protein-based nanoprobes and image-segmentation-based machine learning to establish a new technology for high-throughput quantifying single-vessel permeability in tumor, named Nano-ISML. Through quantitative analysis of >67,000 individual blood vessels from 32 tumor models, it is systematically revealed that there is great heterogeneity in tumor vascular infiltration. It was found that the percentage of high-permeability vessels in different tumors was more than 13 times and the penetration ability in vessels with the highest permeability is >100 times more than vessels with the lowest permeability, which mainly depended on tumor type and vessel type. The research data show that passive extravasation and transendothelial transport were the dominant mechanisms for high- and low-permeability tumor vessels, respectively. The authors proposed a new strategy for classification of hypertonic and hypotonic tumors based on this mechanism. Based on the research results of tumor vascular permeability mechanism of nanomaterials, the authors developed genetically tailored protein nanoparticles with improved transendothelial transport in low-permeability tumors with the help of nano-ISML-assisted nanodrug design, which improved the therapeutic effect of drug delivery system on tumor models.

In addition to the above two cases that guide design of nanomedicine, it is also a new perspective to build a virtual biological environment (such as tumor environment) to simulate the biological behavior of nanodrugs or nanomaterials and then realize optimization of nanodrugs or nanomaterials [119]. However, the effectiveness of practical application of this method should remain for further discussion. On the one hand, the research on the mechanism of nanomaterials is still insufficient, which directly affects the rationality of the constructed virtual environment. On the other hand, it is difficult to guarantee the reliability of physiological parameters of the model.

### 6.5. Simulation Study of Nanodrug Delivery

As a computing method for simulation, application of molecular simulation can intuitively reflect structure and behavior of nanodrugs and obtain dynamic information under different conditions. Although there are still some limitations, simulation of nanodrug delivery can provide sufficient reference information for implementation of an experiment and help to improve design of a nanodrug delivery system. At present, application of molecular simulation in nanodrug delivery mainly focuses on self-assembly, drug release, stability and interaction with cell membrane of nanodrugs. For example, Kordzadeh et al. [120] used molecular dynamics (MD) simulation and density functional theory (DFT) methods to study the delivery, loading and release of doxorubicin using functionalized carbon nanotubes (f-CNT) (functionalized by carboxyl and folic acid, respectively, named CNT-COO and CNT-COO-FA) as carriers. The simulation results showed that, at pH = 7, CNT-COO-FA had stronger binding with drug molecules and could carry more drug. The drug release simulation around the cancer cell model (pH = 5.5, containing folate receptor on cell surface) shows that CNT-COO-FA, with a pH- and ligand-sensitive mechanism, has strong interaction with cancer cells, resulting in higher drug release, which is consistent with the experimental results. Based on the results obtained, it can be found that the pH and ligand sensitivity mechanisms are the reasons for the higher drug delivery efficiency of CNT-COO-FA. These results have certain guiding significance for design of high-performance drug delivery systems.

Moreover, Katiyar et al. [121] studied the effect of media–drug and carrier–drug interactions on pH-responsive drug carriers (using poly(acrylic acid)(PAA) oligomers) through two sets of simulations with molecular dynamics simulation. The first set of simulation results showed that PAA was relatively less ionized in the gastric juice model and formed aggregates because the pH value of gastric fluid was significantly lower than the pH value of intestinal fluid. The second group of simulation results showed that, with an increase in pH, although PAA aggregation decreased, the diffusion coefficient of DOX would decrease due to an increase in the ionic complexation of PAA with DOX. The research results of these two groups show that carrier aggregation and carrier drug interaction are competitive influences that jointly determine drug release of pH-responsive polymers. The method in this paper can also be applied to other new polymers and drugs, which is helpful to design and develop potential drug delivery systems. In short, as an auxiliary tool, molecular simulation can accelerate development of drug delivery systems to a certain extent, but it cannot completely rely on molecular simulation. The limitations of simulation methods make the results inevitably different from the actual situation. More work needs to be completed to explore these problems.

Overall, application space of computing in R&D of nanodrugs is very broad. For complex applications, model prediction and molecular simulation are often used in combination at the same time, which greatly promotes the development process of nanodrug discovery. As an efficient and practical tool, computing has been widely used in development of common drugs. Therefore, its application prospect in nanodrugs can learn from common drugs. However, application of computing in these areas also has problems faced by the above several computing topics, such as insufficient learning ability of algorithms and lack of appropriate descriptors.

## 7. Conclusions

With progress in science and technology, development of computer technology has entered a fast and new era. Significant improvement in computer performance and the availability of big data have greatly promoted development of various algorithms and simulations and further accelerated application of computer technology in artificial intelligence, life science, materials science and other fields. Although drug R&D is a relatively traditional and conservative field, the proportion of computing in drug development has risen rapidly in the past decade. At present, computer technology has been successfully applied in most aspects of the early stage of drug discovery, which has greatly promoted development of drug R&D.

As innovative drugs developed by using nanotechnology, the research and development process of nanodrugs also widely uses computing methods, such as model prediction and molecular simulation, including high-throughput screening, structural characterization, physicochemical and biological properties prediction, pharmacokinetic analysis and toxicity prediction. In the computing tools, application of ML algorithms in AI is dominant and has achieved good research results, which is encouraging.

Although computational research can never replace laboratory experiments, application of computational tools can reduce blindness and contingency in nanodrug development, accelerate development progress and save time and costs. As a technology-intensive and fast-updating discipline, computer technology is bound to be deeply integrated with nanodrug R&D and become the engine to promote high-quality development of nanodrugs.

Despite having achieved many successes in regulation and design of nanodrug applications, computing still faces many challenges. In particular, obtaining a large amount of reliable data remains the main challenge in the process of nanodrug R&D. Compared with small molecular databases (such as PubChem database, containing 111 million compounds), protein databases (such as uniport database, containing more than 200 million protein data) and gene databases (such as KEGG database, containing more than 38 million biological gene data), the amount of data in the nano-field is small and the quality of data is not ideal. On the one hand, credibility of data is highly dependent on the experimental conditions and technical level of the testers. At the same time, the biological system of nanodrugs is extremely complex, which also has a great impact on the instability of the experimental results. On the other hand, the experimental methods related to nanomaterials are expensive, time-consuming and laborious. It is difficult to determine the properties of nanomaterials with a wide variety and large number by experimental methods. One way to solve this dilemma is to build open databases to share data. In addition, nanomaterial data should also be obtained through experimental methods or data mining, and these large amounts of nanomaterial data should be unified into a standard database to facilitate data sharing between different research groups. However, there are still some difficulties in this process and there is a long way to go.

Lack of appropriate descriptors is also a problem we have to face. Construction of nanodrug prediction models often requires participation of nano-descriptors. Appropriate nano-descriptors can greatly improve prediction accuracy. Nano-descriptors used in previous studies are mainly divided into experimental descriptors, ligand descriptors and quantitative calculation descriptors [122]. However, experimental descriptors are time-consuming and laborious and the repeatability is poor. Ligand description is only calculated from the surface ligand and cannot fully include information of nanomaterials, such as type, particle size, ligand distribution and density. Quantitative computing descriptors consume a great deal of computing resources, and it is difficult to calculate nanomaterials with large particle size. Therefore, the lack of descriptors that can characterize the overall structure of nanomaterials is an important obstacle to use of models to predict the biological effects of nanodrugs. One way of thinking is to solve the descriptor problem by using images directly converted from nanostructures. For example, inspired by face recognition technology, researchers have developed a new nanostructure labeling method that automatically converts nanostructures into images and performs convolutional neural network modeling [122]. Compared with the traditional ML method, this kind of method does not need complex nano-descriptors calculation and directly learns nanostructure features from NM images. However, there are still defects in the current nanostructure image and convolution neural network modeling. For example, a nanostructure image shows diversity and difference in nanostructures, but, because of the complexity of nanostructures, it still cannot fully reflect the experimental value. In the future, more advanced 3D or higher-dimensional nanostructure imaging methods are needed [122]. In addition, through molecular dynamics simulation of nanostructures, simulation-derived descriptors from computational simulation can also be regarded as a solution.

Another major challenge is the limitation of molecular simulation for complex systems. Although molecular simulation based on force field is increasingly widely used in chemistry, materials and life systems, it plays an important role in obtaining the structure and dynamic properties of complex molecular systems. However, in application of complex systems, such as nanodrugs, molecular simulation still has bottlenecks, such as limited simulation scale, low accuracy, slow calculation and unstable convergence, that need to be solved. In recent years, development of deep learning methods and software and hardware architectures has provided new technical support and ideas for progress in molecular simulation. The precision of molecular dynamics simulation based on neural network can be comparable with that based on quantum mechanics, but its operation speed is tens of thousands of times faster than that of the latter. In general, achieving higher efficiency, simulating larger systems, achieving longer evolution time and obtaining more accurate simulation results are the future development trends of molecular simulation in application of nanomedicine.

## 8. Future Perspectives

With the rapid development of computer technology and nanotechnology, many of the above challenges are likely to be solved in the near future. Application of computing in the process of nanodrug research and development is bound to be more extensive and deeper, which will also ameliorate the situation of high cost and high risk of failure in R&D of nanodrugs. Similarly, nanodrugs will also have better ADMET characteristics, targeting activity and safety under precise regulation and design. With the help of computer technology, we will also have a deeper understanding of the biological effects and mechanism of action of nanodrugs and provide more accurate treatment plans for more clinical treatment groups. In addition, calculation will not only be limited to the development stage of nanodrugs but also applied to recruitment of patients in clinical trials. For a long time, it has been time-consuming and laborious to recruit enough suitable patients for clinical trials. Building models and forecasting disease data may help pharmaceutical companies identify and recruit target patients more accurately and quickly.

On the other hand, popular AI technology can also further strengthen cooperation with nanodrug R&D. For example, computer vision technology of AI can be used to analyze the distribution imaging and tissue imaging of nanodrugs or nanomaterials in vivo, and it is more likely to find information that is ignored by human beings. In addition, with increasing research on nanodrugs, several related research papers have also been published. How to retrieve high-quality nanodrug literature that meets requirements in a short time and quickly find more hidden information from the intricate literature is still a major problem for researchers. Further, text mining based on natural language processing technology of AI is an effective way to solve the above problems.

In short, computing has shown potential in all stages of nanodrug R&D. In addition to the content discussed above, there is more room for computing to be explored. However, development of nanodrugs is still a slow business and application of computing in it is just beginning. We are unlikely to see earth-shaking changes overnight. However, there is no doubt that computing will change some processes in nanodrugs R&D and accelerate progress.

## Figures and Tables

**Figure 1 pharmaceutics-15-01064-f001:**
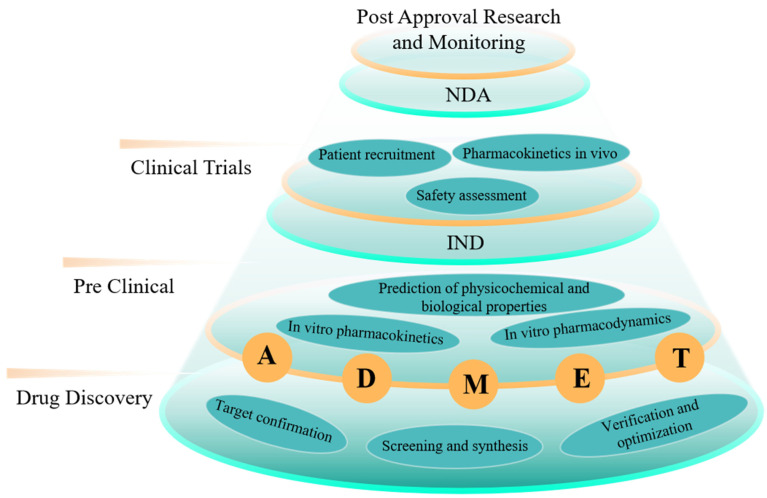
A brief introduction to research and development process of nanodrugs and applications in computing.

**Figure 2 pharmaceutics-15-01064-f002:**
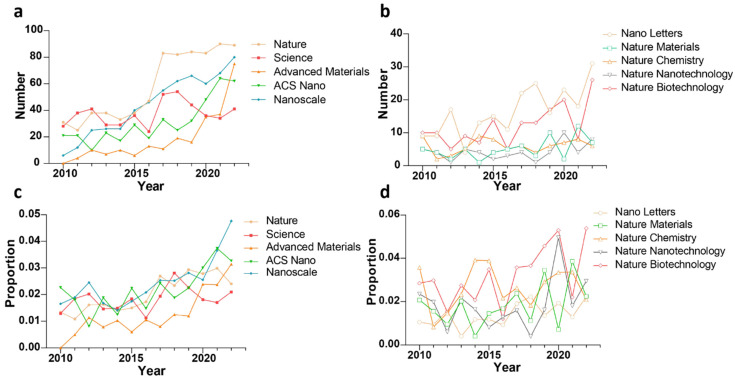
Number and proportion of articles related to computing published in important journals and journals in nano-field between 2010 and 2022. (**a**,**b**) Changes in the number of computing-related articles published from 2010 to 2022 in 10 journals. (**c**,**d**) Changes in the proportion of computing-related articles in total published articles from 2010 to 2022 in 10 journals.

**Figure 3 pharmaceutics-15-01064-f003:**
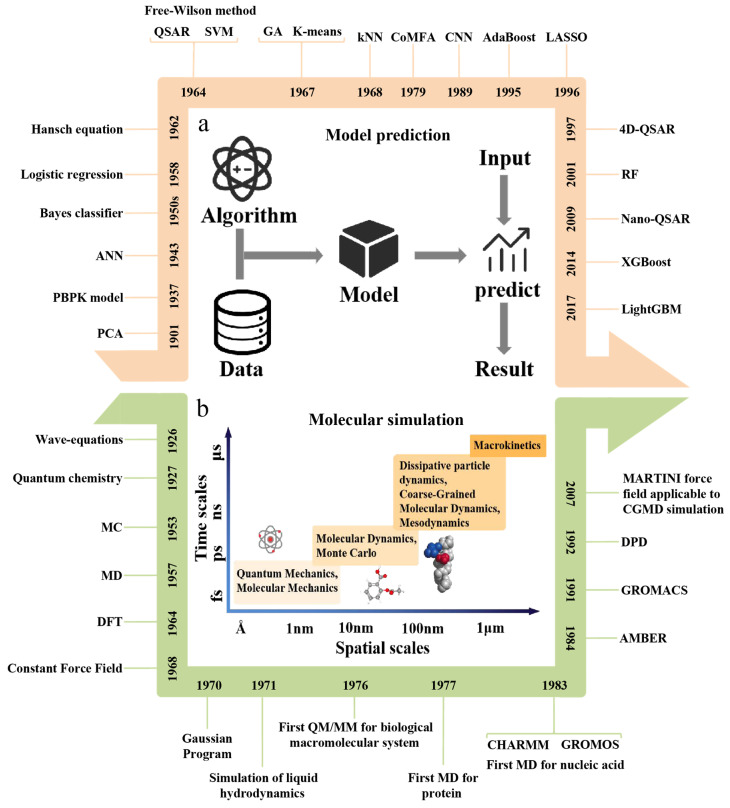
Timeline of relevant developments to predict and simulate the physicochemical properties and biological effects of nanomaterials. (**a**) Schematic of model prediction workflow. (**b**) Categories of molecular simulation methods. AdaBoost: adaptive boosting, ANN: artificial neural network, CGMD: coarse-grained molecular dynamics, CNN: convolutional neural network, CoMFA: comparative molecular field analysis, DFT: density functional theory, DPD: dissipative particle dynamics, GA: genetic algorithm, kNN: k-nearest neighbor, LASSO: least absolute shrinkage and selection operator, LightGBM: light gradient boosting machine, MC: Monte Carlo, MD: molecular dynamic, MM: molecular mechanic, PBPK: physiologically based pharmacokinetic, PCA: principal components analysis, QM: quantum mechanics, QSAR: quantitative structure–activity relationship, RF: random forest, SVM: support vector machine, XGBoost: eXtreme gradient boosting.

**Figure 4 pharmaceutics-15-01064-f004:**
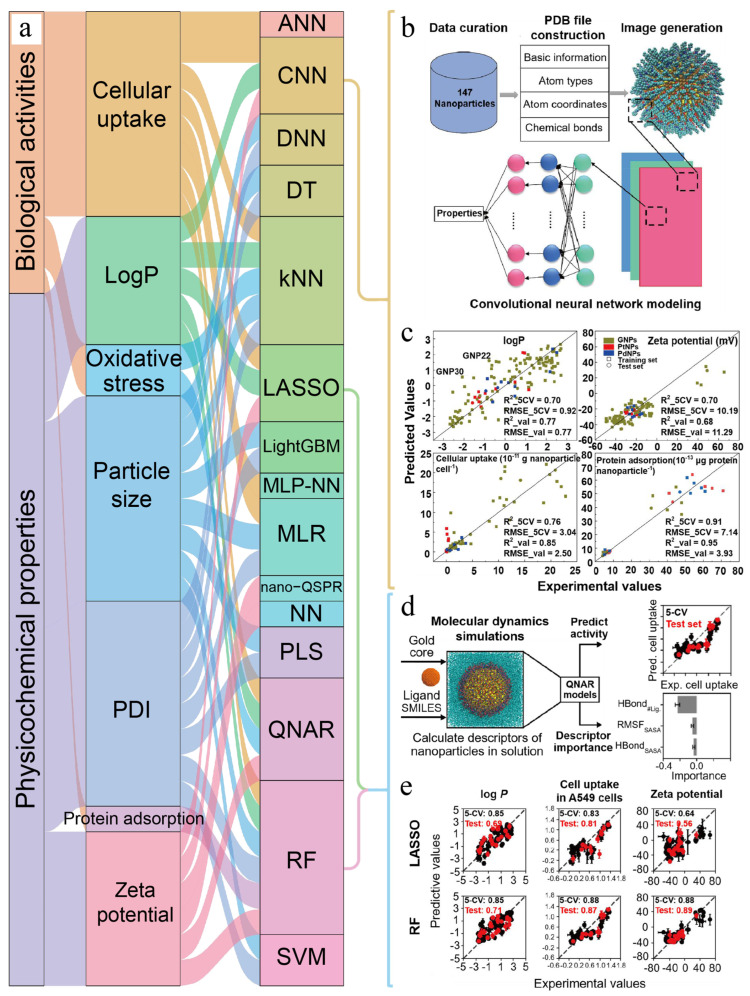
Application of computing in predicting physicochemical properties and biological activities of nanomaterials. (**a**) The second column represents the specific properties and the third column represents the algorithms and models [14,24,28,29,30,31,32,33,34,35,36]. Each rectangle represents an item, and the size of the rectangle is directly proportional to the connection degree of each item. (**b**,**c**) Schematic of the computational workflow and LeNet convolutional neural network modeling results. Adapted with permission from [28], American Chemical Society, 2020. (**d**,**e**) Workflow diagram of prediction model and prediction accuracy of QNAR models. Adapted with permission from [14], American Chemical Society, 2022. LogP: adaptive boosting, PDI: polymer dispersity index, ANN: artificial neural network, CNN: convolutional neural network, DNN: deep neural network, DT: decision table, kNN: k-nearest neighbor, LASSO: least absolute shrinkage and selection operator, LightGBM: light gradient boosting machine, MLR: multiple linear regression, MLP-NN: multilayered perceptron neural network, QSAR: quantitative structure–activity relationship, NN: neural network, PLS: partial least square, QNAR: quantitative nano-structure activity relationships, RF: random forest, SVM: support vector machine.

**Figure 5 pharmaceutics-15-01064-f005:**
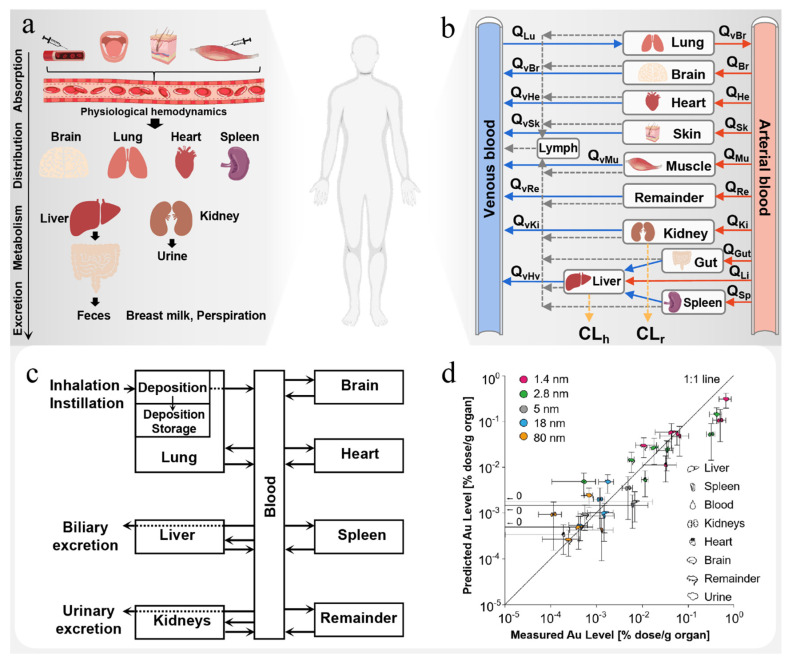
ADME process in vivo and PBPK model of nanodrugs. (**a**) ADME of nanoparticles in vivo. Adapted with permission from [53], Elsevier, 2020. (**b**) Schematic diagram of PBPK model structures for nanoparticles. (**c**) Schematic diagram of the AuNP PBPK model. (**d**) Comparison between simulation results and experimental results of the AuNP biodistribution. Adapted with permission from [54], Springer, 2015.

**Figure 6 pharmaceutics-15-01064-f006:**
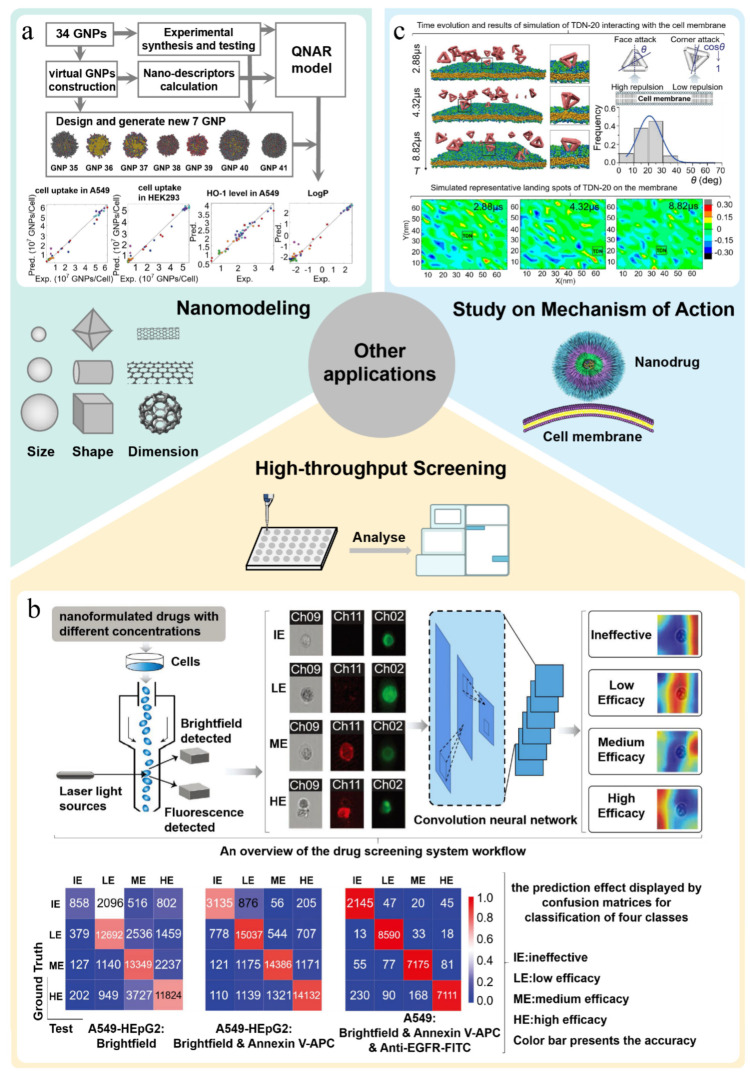
Computational applications in other aspects of nanodrug development. (**a**) Schematic workflow of predictive modeling and QNAR model performance. Adapted with permission from [24], American Chemical Society, 2017. (**b**) A drug screening system utilizing convolutional neural network based on flow cytometry single-cell images provides accurate and rapid approach for screening drug and nano-carrier drug system. Adapted with permission from [113], Wiley Online Library, 2021. (**c**) Dissipative particle dynamics simulations of TDN-20 interacting with the cell membrane and result of the TDNs attacking the membrane. Adapted with permission from [114], American Chemical Society, 2018.

## Data Availability

The data presented in this study are available on request from the corresponding author.

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
