# Peer review of "Application of Computing as a High-Practicability and -Efficiency Auxiliary Tool in Nanodrugs Discovery"

_pharmaceutics, 2023, doi:10.3390/pharmaceutics15041064_

Round 1

Reviewer 1 Report

This is an interesting and timely review article about the use of computational technology to predict the pharmaceutical properties of naondrugs. The review is very thorough, well organized, and clearly written. In terms of suggestions, artificial intelligence is now making headlines in the media so it would be worth considering the extent to which it has been applied to nanodrugs

Author Response

Thank you very much for your comments. These suggestions are very helpful for us to improve the quality of the manuscript. We have revised the manuscript in accordance with the suggestions. We have carefully checked the entire manuscript for typographic, grammatical and formatting. And the important changes made in the revised manuscript have been highlighted in yellow.

We greatly appreciate for the valuable comments from you. Moreover, we would like to express our thanks for your processing the reviewing processes. We hope the revised manuscript will be suitable for publication in Pharmaceutics.

Reviewer 2 Report

Referee Report

Title: Computing: High practicability and efficiency auxiliary tool in drug discovery

Manuscript ID: pharmaceutics-2283308

By Xu et al

Submitted to Pharmaceutics (ISSN 1999-4923)

Comments

This is a topical review on computing tools used in drug discovery. I have the following comments regarding this work:

1.       Title: From the title, I cannot see this is a review article. I thought it would be a research paper about developing a new computing tool in drug discovery. Please rephrase the title to let readers know it is a review article.

2.       Abstract: It is very important to state the intention of the authors to conduct such a review. Why the authors write such review?

3.       Introduction: Nanodrugs can be molecules but not only nanoscale particles (nanoparticles).

4.       It is good to enclose all figures in the text instead of attached them at the end of the article.

5.       The authors focused very much on predicting the properties of nanodrug. Did they have any information about the design of the nanodrug such as functionalized nanodrug for theranostics?

6.       Can the authors provide information about the simulation of drug delivery?

7.       Section 2-6: The authors should provide a paragraph in the end to summarize what are the progresses and what are still missing in those computing topics.

8.       The authors can consider to separate Section 7 into two Sections of Future Perspectives and Conclusions.

Author Response

(The authors gave the same response as above.)

Round 2

Reviewer 2 Report

The authors did a great job to modify their manuscripts as per my comments. Their authors responses and revised manuscript are very good. The quality and presentation of this work are improved.